

# Effect of post COVID-19 on body composition, physical fitness, sleep quality and quality of life among young adults: a cross-sectional study of matched pairs

Phatcharawadee Srirug[1,2], Chatkaew Pongmala[3], Balkis Mayeedeng[1], Nurulfatin Yusoh[1], Sofiya Malee[1] and Chadayu Udom[1,2]

[1] Department of Physical Therapy, School of Allied Health Sciences, Walailak University, Thasala, Nakhonsrithammarat, Thailand
[2] Walailak University, Movement Science and Exercise Research Center, Nakhon Sri Thammarat, Thailand
[3] Functional Neuroimaging, Cognitive, and Mobility Laboratory, Department of Radiology, University of Michigan, Michigan, United States

Corresponding author
Chadayu Udom,
uchadayu@mail.wu.ac.th

## ABSTRACT

**Background:** Coronavirus 2019 (COVID-19) causes lung tissue inflammation, affects mental health, and disturbs sleep and the musculoskeletal system. This study aimed to investigate the effects of COVID-19 on physical function and quality of life by comparing the body composition, physical fitness, sleep quality and quality of life between Thai young adults with and without post COVID-19 infection.

**Methods:** A cross-sectional study was comprised of two matched groups (post COVID-19 group and non COVID-19 group) with 36 participants in each group. Data about body composition, physical fitness, sleep quality and quality of life were collected using the bioelectrical impedance analysis, the 1-min sit-to-stand test, the hand grip strength test, the Pittsburgh sleep quality questionnaire and the EuroQol-5D-5L, respectively. Independent samples T-test, Mann-Whitney U test and Chi-square test were used to compare between the two groups.

**Results:** The changes in oxygen saturation and respiratory rate after the physical fitness test and the sleep quality analysis showed a statistically significant difference between the groups with and without post COVID-19 infection ($p = 0.006$, $p = 0.003$ and $p = 0.003$, respectively). However, quality of life and body composition were not significantly different between groups.

**Conclusions:** COVID-19 influenced the changes in oxygen saturation and respiratory rate after the physical fitness test and the sleep quality analysis in young adults. The results should be utilized to facilitate physical rehabilitation for COVID-19-infected individuals following infection. Those who have not been infected with COVID-19 must be informed of self-protection measures to avoid contracting the virus.

## INTRODUCTION

Coronavirus disease 2019 (COVID-19) is a disease caused by exposure to the severe acute respiratory syndrome coronavirus 2 (SAR-COV-2), which began to spread in China since 2019. The number of infections increased in many countries. Therefore, the World Health Organization (WHO) declared an epidemic in March 2020. As of April 13, 2024, more than 704 million people worldwide have been infected and more than 7 million have died from COVID-19. Thailand has the 33rd highest number of infected people in the world, with a total of more than 4 million infected and more than 30,000 people who have died from COVID-19 (*Worldometers, 2024*).

When the body is infected with SAR-COV-2, it causes lung tissue inflammation. These include interstitial pneumonia and respiratory distress syndrome, which cause organs such as the heart, lungs, brain and nervous system to fail (*Zhou et al., 2020*). Furthermore, it affects mental health (anxiety and depression), disturbs sleep and the musculoskeletal system (pain and muscle weakness), and causes fatigue (*WHO, 2020*).

Recent studies have shown that some patients may develop complications from infection, and 11 to 24 percent of patients' symptoms may last longer than 3 months after exposure (*Cirulli et al., 2020*; *Ding et al., 2020*; *WHO, 2020*). Therefore, it may reduce physical fitness and quality of life. In addition, after the COVID-19 infection has disappeared, patients may still have other symptoms. Symptoms are usually found 4 weeks or more after being infected, also known as the long Covid condition (*Ladds et al., 2020*). This could be a new symptom or a residual effect after treatment; and symptoms such as hypoxia, shortness of breath or dyspnea, decreased ability to work (reduced ability to work) (*Bryson, 2021*; *Santus et al., 2020*), and fatigue (*Shanbehzadeh et al., 2021*) may persist for more than or equal to 12 weeks. Factors affected by fatigue consist of conditional factors and physiological factors. The conditional factors include job characteristics, environment, and individual physical and mental abilities. The physiological factors include central factors (levels of neurotransmitters, neuron activation, inflammation, and multiple sclerosis), psychological factors (stress, anxiety, depression, and fear) and peripheral factors (skeletal muscle) (*Rudroff et al., 2020*).

Previous literature have reported that physical fitness and quality of life were studied in patients with COVID-19, but very few studied the effects of post COVID-19 infection (*Rudroff et al., 2020*). Factors affecting the symptoms of post COVID-19 patients include high blood pressure, obesity, mental health condition (*Tenforde et al., 2020*), age, associated comorbidities, severity of COVID-19 symptoms, and severity of COVID-19 infection (*Sharma, Bharti & Garg, 2022*). Therefore, this study aimed to examine the impact of COVID-19 infection on body composition, physical fitness and quality of life in Walailak University students who had recovered from COVID-19 infection in order to provide a guideline to raise awareness on preventative measures and physical rehabilitation after COVID-19 infection.

## MATERIALS AND METHODS

### Participants and setting

This study is a cross-sectional study (Observational study: matched case-control). Data was collected from July to September 2022 at the Physical Therapy Research Laboratory, Walailak University. Walailak University granted ethical approval to carry out the study within its facilities (Ethical Application Ref: WUEC-22-204-01).

The study objectives, research process, and benefits to be gained from participating in the study were explained to all volunteers. Participants signed a consent form to participate in the study, and selection of participants was according to the inclusion and exclusion criteria. Data was collected from the research project participants who met the inclusion criteria (Fig. 1).

The volunteers interested in participating in the study were screened, and those who could not understand the questionnaire and had peripheral blood oxygen saturation ($SpO_2$) <95% were excluded from the study. The inclusion criteria of each group were as follows:

Post COVID-19 group: This group included Walailak University students, both male and female with normal range body mass index (BMI) from 18.5 to 22.9 kg/m$^2$ (*Supadanaison, Panklang & Seechumsang, 2019*), who had been diagnosed with COVID-19 and had recovered or completed the quarantine period within 1–3 months after being infected. At least one symptom should remain. The average number of days since the COVID-19 infection was 50 days and ranged from 30 to 119 days.

Non COVID-19 group: This group included Walailak University students, both male and female with normal range BMI, who have never been diagnosed with COVID-19 infection.

### Matched-pair analysis

Each analyzed case of the post COVID-19 group ($n = 36$) was matched with one non COVID-19 group ($n = 36$) applying matching criteria in the following prioritization order: gender, age and physical activity level. Because of these confounding factors, there was an effect on muscle integrity and may have a final effect on the body composition, grip strength, the 1-minute sit-to-stand test (1MSTS) and fatigue score.

### Data collection

This study collected data in two parts. The first part was comprised of data collected by four questionnaires as follows: 1) general characteristics (including age, gender, weight, height, body mass index, waist circumference, physical activity level, smoking habit, average sleep duration within the past 1 year, behavior according to protective measures, number of vaccines received, and quarantine as a high-risk group); 2) the European Quality of Life five Dimensions five Level questionnaire (EuroQol-5D-5L) (Registration No. 49009), which has a good level of structural validity (*Kimman et al., 2013*). The score was classified into two levels: perfect health and not perfect health. Perfect health was defined with a score of one on each dimension, and not perfect health was defined as any other scores in one or more of the five dimensions; and 3) the Pittsburgh Sleep Quality
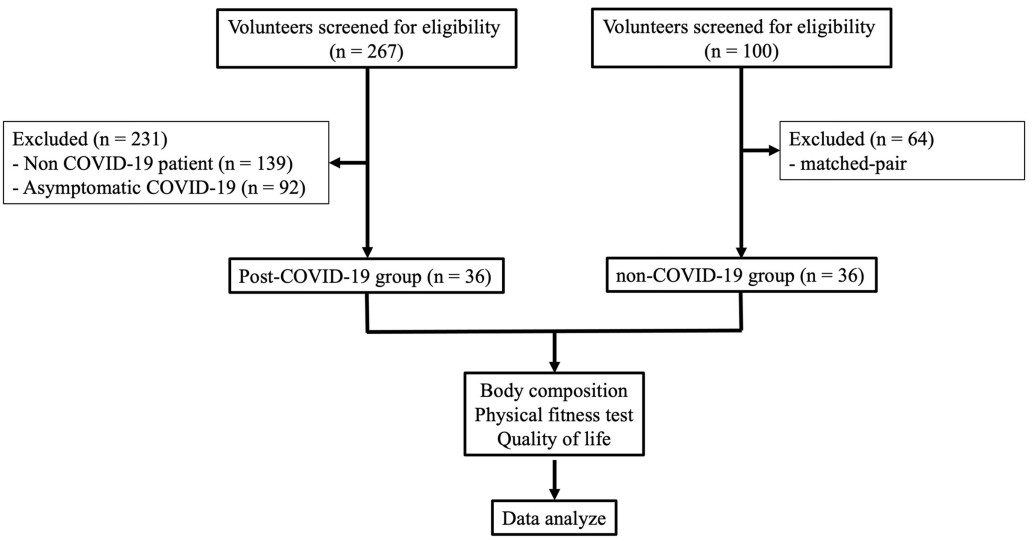

Figure 1 Flow chart of participant recruitment.

questionnaire—Thai version (Pittsburgh sleep quality index score; PSQI score), which had a Cronbach's alpha of 0.84, intraclass correlation coefficient of 0.89, sensitivity of 77.78%, specificity of 93.33%, and could be used to classify people with sleep problems (*Sitasuwan et al., 2014*). The score was classified into two levels: good sleep quality and poor sleep quality. Good sleep quality referred to a PSQI score of five or less, and poor sleep quality referred to a PSQI score greater than five. The second part was comprised of data collected by using the following: 1) the bioelectrical impedance analysis (BIA) (Tanita SC-330 Body Composition Analyzer; Tanita Corp., Tokyo, Japan) to assess body composition including body fat, muscle mass, visceral fat level (range from 1–59; 1–12 indicates a healthy level of visceral fat and 13–59 indicates an excess level of visceral fat), lean body mass and total body water; and 2) the physical fitness test including the 1-min sit-to-stand test (1MSTS) and the hand grip strength test, which was measured using a hand grip dynamometer (Takei TKK5001; Takei Scientific Instruments Co. Ltd, Tokyo, Japan) (*Zaccagni et al., 2020*). For hand grip strength, the participants were asked to use their dominant hand to squeeze the hand grip dynamometer as hard as they could and hold for 5 s, repeat three times, and rest 1 min each time; and the highest value was used. The 1MSTS (*Bohannon & Crouch, 2019*), with minimal important difference of three and ICC of 0.99, was used for the assessment of exercise capacity, which would be equivalent to the 6-min walk test (6MWT). The testing method was as follows: the participants were seated on armless and wall-mounted chairs, had arms crossed over chest, and asked to stand up and sit down for one round before starting the test. Blood pressure (mmHg), heart rate (bpm), respiratory rate (bpm), oxygen saturation ($SpO_2$) using pulse oximeter, and fatigue level using Borg CR-10 were assessed before and after the 1MSTS. The blood pressure values were represented as mean arterial pressure (MAP). To determine the MAP, double the diastolic blood pressure, add this value to the systolic blood pressure, and then divide the total by three. During the test, the change in blood oxygen level was assessed for 1 min over the

course of the test. The differences in heart rate, respiratory rate, $SpO_2$ and fatigue score were calculated by subtracting the values before the test from those after the test.

## Sample size calculation

The sample size for comparison between the post COVID-19 and non COVID-19 groups was calculated using the following formula for group comparison with an independent t-test (*Wang & Ji, 2020*):

$$n = \frac{2\sigma^2 \left(Z_{1-\alpha/2} + Z_{1-\beta}\right)^2}{(\mu1 - \mu2)^2}$$

Based on the number of repetition of 1MSTS, $Z_{1-\alpha/2} = 1.96$, $Z_{1-\beta} = 1.645$, $\mu1 - \mu2 = 6.7$, and $\sigma^2 = 40.96$ (*Peroy-Badal et al., 2024*), the required sample size was calculated to be 29. In considering 25% drop out, the required sample size was increased to 36 persons per group. When calculating the effect size using G*power, it was found to be 0.51, indicating that the magnitude of the outcome difference is moderate between the post-COVID-19 patients and the non-COVID-19 group.

## Statistical analysis

Data analysis was performed using SPSS Version 26.0, with statistical significance set at $p < 0.05$. There were no missing data for any parameters. Descriptive statistics were used to analyze the general characteristics of the participants, including age, gender, BMI, and physical activity level. The Shapiro-Wilk test was employed to assess the normality of all variables. For comparing body composition, physical fitness, and quality of life between groups, the Mann-Whitney U test was used for non-normally distributed data, while the independent t-test was used for normally distributed data. For comparing sleep quality (categorical data), the Chi-square test was used.

## RESULTS

Out of a total of 267 volunteers, 128 had been exposed to COVID-19, representing 47.9%. Out of the 128 that had been exposed, 36 post COVID-19 patients (28.1%) had passed the inclusion criteria, whereas 92 were excluded due to asymptomatic COVID-19. Of the 72 participants in the study, 36 had never been infected with COVID-19 and 36 had been infected with COVID-19. Most of the participants were female (80.6%), with light physical activity levels (75%), and non-smokers (97.2%). In addition, 69.4% of the post COVID-19 group and 63.9% of the non COVID-19 group did not get enough sleep during the night. Regarding quarantine as a high-risk group, 75% of the post COVID-19 group and 58.3% of the non COVID-19 group had been quarantined as a high-risk group. Additionally, there was no statistically significant difference between the two groups, as shown in Table 1.

## General characteristics of the post COVID-19 participants (*n* = 36)

From the group of post COVID-19 participants, 88.9% had been infected only 1 time, and 52.8% didn't know the cause of infection. While infected with COVID-19, the majority (80.6%) experienced only mild symptoms, with the highest number of symptoms at 12;

**Table 1 General characteristics of the participants in the post COVID-19 (n = 36) and non COVID-19 (n = 36) groups.**

| General characteristics | Non COVID-19 (n = 36) | | | Post COVID-19 (n = 36) | | | p-value |
|---|---|---|---|---|---|---|---|
| | Mean ± SD | Range | n (%) | Mean ± SD | Range | n (%) | |
| Age (yrs) | 19.58 ± 1.10 | 18–21 | | 19.62 ± 1.09 | 18–21 | | 0.811 |
| **Gender** | | | | | | | 1.000 |
| – Male | | | 7 (19.4) | | | 7 (19.4) | |
| – Female | | | 29 (80.6) | | | 29 (80.6) | |
| BMI (kg/m$^2$) | 20.53 ± 1.38 | 18.16–22.75 | | 20.04 ± 1.55 | 18.00–22.74 | | 0.884 |
| Waist circumference (cm) | 68.75 ± 4.43 | 60–82 | | 68.73 ± 4.26 | 60–82 | | 0.978 |
| **Physical activity level** | | | | | | | 0.810 |
| Light | | | 27 (75.0) | | | 27 (75.0) | |
| Moderate | | | 6 (16.7) | | | 6 (16.7) | |
| High | | | 3 (8.3) | | | 3 (8.3) | |
| Sleep duration during the past year (hrs) | 6.39 ± 1.10 | 5–10 | | 6.43 ± 1.13 | 4–10 | | 0.637 |

| General characteristics | n (%) | | p-value |
|---|---|---|---|
| | Non COVID-19 (n = 36) | Post COVID-19 (n = 36) | |
| **Sleep quality** | | | 0.620 |
| Adequate | 13 (36.1) | 11 (30.6) | |
| Not adequate | 23 (63.9) | 25 (69.4) | |
| **Smoking behaviors** | | | 0.984 |
| Non-smoker | 35 (97.2) | 35 (97.2) | |
| Smoker | 1 (2.8) | 1 (2.8) | |
| **Secondary smoker** | | | 0.227 |
| Yes | 16 (44.4) | 11 (30.6) | |
| No | 20 (55.6) | 25 (69.4) | |
| **Understand COVID-19** | | | 0.317 |
| Yes | 36 (100) | 35 (97.2) | |
| No | 0 (0) | 1 (2.8) | |
| **Practice following guideline for COVID-19** | | | 0.055 |
| Sometimes | 18 (50.0) | 10 (27.8) | |
| Strictly | 18 (50.0) | 26 (72.2) | |
| **Number of vaccines received** | | | 0.317 |
| 0 | 0 (0) | 1 (2.8) | |
| 2 | 22 (61.1) | 17 (47.2) | |
| 3 | 10 (27.8) | 15 (41.7) | |
| 4 | 4 (11.1) | 3 (8.3) | |
| **Quarantine as a high-risk group** | | | 0.136 |
| Yes | 21 (58.3) | 27 (75.0) | |
| No | 15 (41.7) | 9 (25.0) | |

Note:
BMI, body mass index.

**Table 2 Characteristics of the post COVID-19 participants (*n* = 36).**

| Characteristics | *n* (%) |
|---|---|
| **How many times you have been infected with COVID-19** | |
| 1 time | 32 (88.9) |
| 2 times | 4 (11.1) |
| **Do you know the cause of COVID-19 infection** | |
| Yes | 17 (47.2) |
| No | 19 (52.8) |
| **Drug received during COVID-19 infection** | |
| Symptomatic medication | 26 (72.2) |
| Favipiravir with symptomatic medication | 10 (27.8) |
| **Place of treatment during COVID-19 infection** | |
| Home quarantine | 9 (25.0) |
| Quarantine in the community or university | 8 (22.2) |
| Hospital | 1 (2.8) |
| Hospital | 18 (50.5) |
| **How many symptoms you have during COVID-19 infection** | |
| 3 | 1 (2.8) |
| 4 | 3 (8.3) |
| 5 | 8 (22.2) |
| 6 | 5 (13.9) |
| 7 | 2 (5.6) |
| 8 | 4 (11.1) |
| 9 | 5 (13.9) |
| 10 | 2 (5.6) |
| 11 | 4 (11.1) |
| 12 | 2 (5.6) |
| **Symptom during COVID-19 infection** | |
| Sore throat | 35 (97.2) |
| Cough | 33 (91.7) |
| Fever | 30 (83.8) |
| Chest pain | 9 (25.0) |
| Dyspnea | 12 (33.3) |
| Hair loss | 15 (41.7) |
| Fatigue | 27 (75.0) |
| Loss of taste | 10 (27.8) |
| Loss of smell | 7 (19.4) |
| Headache | 23 (63.9) |
| Nausea/Vomiting | 8 (22.2) |
| Diarrhea | 11 (30.6) |
| Insomnia | 12 (33.3) |
| Muscle ache | 23 (63.9) |
| Confusion | 5 (13.9) |
| Memory problem | 4 (11.1) |
| Runny or stuffy nose | 1 (2.8) |

**Table 3 Comparison of body composition of the post COVID-19 ($n$ = 36) and non COVID-19 ($n$ = 36) groups.**

| Body composition | Non COVID-19 ($n$ = 36) | | Post COVID-19 ($n$ = 36) | | $p$-value |
|---|---|---|---|---|---|
| | Mean ± SD | Median (Quartile 1–3) | Mean ± SD | Median (Quartile 1–3) | |
| Body fat (%) | 22.28 ± 5.75 | 24.65 (19.37–27.35) | 23.02 ± 6.26 | 24.75 (20.02–27.12) | 0.875[a] |
| Muscle mass (kg) | 37.85 ± 4.91 | 36.55 (34.62–39.10) | 38.21 ± 5.34 | 36.85 (34.87–40.02) | 0.604[a] |
| Visceral fat level | – | 2 (1–3) | – | 2 (1–3) | 0.725[a] |
| Lean body mass (kg) | 37.08 ± 4.55 | 36.40 (34.04–37.97) | 37.42 ± 4.88 | 36.41 (34.97–38.55) | 0.569[a] |
| Total body water (%) | 51.83 ± 3.52 | 51.29 (48.95–53.14) | 58.42 ± 17.02 | 52.09 (49.09–57.69) | 0.265[a] |

**Note:**
[a] Mann-Whitney U test was used.

and the most common symptom was sore throat followed by cough and fever, as shown in Table 2.

### Body composition between the post COVID-19 ($n$ = 36) and non COVID-19 ($n$ = 36) groups

Body composition composed of body fat, muscle mass, visceral fat levels, lean body mass, and total body water. It was found that there was no statistically significant difference in all parameters of the body composition between the post COVID-19 and non COVID-19 groups, as shown in Table 3.

### Physical fitness between the post COVID-19 ($n$ = 36) and non COVID-19 ($n$ = 36) groups

From the physical fitness test evaluating the strength of the hand muscles and assessing the 1MSTS, changes were detected in the heart rate, respiratory rate, blood oxygen saturation, and blood pressure pre and post 1MSTS test.

When compared between the post COVID-19 and non COVID-19 groups, the change of respiratory rate and blood oxygen saturation exhibited a statistically significant difference ($p$ = 0.003 and $p$ = 0.006, respectively), as shown in Table 4.

### Sleep quality between the post COVID-19 ($n$ = 36) and non COVID-19 ($n$ = 36) groups

A statistically significant difference was found between the post-COVID-19 and non-COVID-19 groups in terms of sleep quality. The non COVID-19 group had a similar proportion of people with good and poor sleep quality, but the majority of the post-COVID-19 participants had poor sleep quality (80.6%), as indicated in Table 5.

### Quality of life between the post COVID-19 ($n$ = 36) and non COVID-19 ($n$ = 36) groups

When quality of life was compared between the post-COVID-19 and non COVID-19 groups, there was no statistically significant difference (Table 6).

**Table 4 Comparison of physical fitness between the post COVID-19 (*n* = 36) and non COVID-19 (*n* = 36) groups.**

| Physical fitness | Non COVID-19 (*n* = 36) | | Post COVID-19 (*n* = 36) | | *p*-value |
|---|---|---|---|---|---|
| | Mean ± SD | Range/Median (Quartile 1–3) | Mean ± SD | Range/Median (Quartile 1–3) | |
| Grip strength (kg/BW) | 0.51 ± 0.10 | 0.34–0.82 | 0.49 ± 0.10 | 0.31–0.77 | 0.528[b] |
| Δ Heart rate (bpm) | 46.11 ± 14.58 | 22–90 | 40.53 ± 12.70 | 11–65 | 0.121[b] |
| MAP before 1MSTS (mmHg) | 73.30 ± 6.94 | 60.00–86.00 | 73.42 ± 9.20 | 52.00–106.66 | 0.913[b] |
| MAP after 1MSTS (mmHg) | 76.39 ± 8.55 | 60.00–96.33 | 76.48 ± 9.50 | 55.33–101.33 | 0.461[b] |
| 1MSTS (time) | 29.31 ± 6.63 | 30 (24.62–34.87) | 29.66 ± 6.49 | 28.75 (25–34.75) | 0.866[a] |
| Δ Respiratory rate (bpm) | 6.42 ± 1.79 | 6 (5–7.75) | 7.06 ± 1.95 | 8 (6.25–9) | 0.003[a*] |
| Δ SpO$_2$ (%) | 0.89 ± 0.70 | 1 (0–1) | 1.15 ± 0.83 | 1 (1–2) | 0.006[a*] |
| Δ Fatigue score | 1.59 ± 1.22 | 1.25 (1–2.37) | 1.43 ± 1.17 | 1 (0.5–2) | 0.238[a] |

Notes:
[*] Significance at $p < 0.05$.
[a] Mann-Whitney U test was used.
[b] Independent samples T-test was used.
MAP, mean arterial pressure; 1MSTS, 1-min sit-to-stand test; SpO2, peripheral blood oxygen saturation.

**Table 5 Comparison of sleep quality between the post COVID-19 (*n* = 36) and non COVID-19 (*n* = 36) groups.**

| Sleep quality | *n* (%) | | *p*-value |
|---|---|---|---|
| | Non COVID-19 (*n* = 36) | Post COVID-19 (*n* = 36) | |
| Good sleep quality (>5) | 19 (52.8) | 7 (19.4) | 0.003[a*] |
| Poor sleep quality (≤5) | 17 (47.2) | 29 (80.6) | |

Notes:
[a] Chi-square test was used.
[*] Significance at $p < 0.05$.

**Table 6 Comparison of quality of life between the post COVID-19 (*n* = 36) and non COVID-19 (*n* = 36) groups.**

| Quality of life | *n* (%) | | *p*-value |
|---|---|---|---|
| | Non COVID-19 (*n* = 36) | Post COVID-19 (*n* = 36) | |
| – Not perfect health | 19 (52.8) | 20 (55.6) | 0.814[a] |
| – Perfect health | 17 (47.2) | 16 (44.4) | |

Note:
[a] Mann-Whitney U test was used.

# DISCUSSION

The purpose of this study was to determine the impact of COVID-19 on body composition, physical fitness, sleep quality and quality of life. There was no statistically significant difference in all parameters of body composition (body fat, muscle mass, visceral fat, lean body mass and total body water) between the post COVID-19 and non COVID-19 groups, contradicting the previous studies that post COVID-19 group had statistically significant increase in body fat (*Atieh et al., 2024*; *Peball et al., 2024*) and

decrease in muscle mass (*Montes-Ibarra et al., 2023*). This might be because the previous study had recruited participants who had been infected for more than 3 months, but the participants of the post COVID-19 group in this study had been infected for less than 3 months (average 50 days). Therefore, that the long duration after infection might have influenced the change in body composition. It might be interesting to follow up with the post COVID-19 group for more than 3 months. Furthermore, this might be attributable to the specific strain of SARS-CoV-2. During the data collection period, the dominant SARS-CoV-2 strains circulating in Thailand were primarily from the Omicron variant, particularly subvariants BA.1 and BA.2. These subvariants generally cause milder illness and less frequent loss of taste and smell compared to earlier variants like Delta (*WHO, 2024*). However, this study did not record the SARS-CoV-2 strain, so future research should identify the strain to better understand its impact on the outcomes.

The number of times to get up and sit for 1 min was not significantly different between the post COVID-19 and non COVID-19 groups. The post COVID-19 group had a greater decrease in blood oxygen saturation ($SpO_2$) after the 1MSTS than the non COVID-19 group. $SpO_2$ was measured using pulse oximetry, which provides a convenient, noninvasive method to measure blood oxygen saturation continuously. It can also help to eliminate medical errors. Pulse oximetry has a sensitivity of 92% and a specificity of 90% (*Hafen & Sharma, 2022*). Among the post COVID-19 group, pneumonia developed during the acute stage of infection (*Peter et al., 2020*) and decreased lung function due to a change in the structure of the lungs. It was found that lung elasticity was reduced by post-inflammatory fibrosis resulting in an uneven distribution of capillaries (*Rana et al., 2021*), causing breathing difficulties so that the exchange of oxygen in the blood was reduced. However, the difference was not clinically significant for detecting exercise-induced desaturation. A decrease in oxygen saturation of three percentage points or more during the test was considered clinically significant for detecting exercise-induced desaturation (*Briand et al., 2018*).

In addition, there was a statistically significant difference in the respiratory rate ($p < 0.05$) after the 1-min sit-to stand test, with those who had post COVID-19 showing a higher change in respiratory rate compared to those who had not been exposed to COVID-19. The respiratory rate is a specific and early indicator of physiological conditions like hypoxia and hypercapnia, highlighting its importance in identifying patient deterioration and prompting timely intervention. This is because dyspnea is a common residual symptom after exposure to COVID-19 (*Nalbandian et al., 2021*), even in mild cases. This was consistent with a previous study of hyperventilation in patients who had been exposed to COVID-19 with mild symptoms, and it was found that patients who had been exposed to COVID-19 had increased breathing rate and decreased ability to exercise during cardiopulmonary function tests (*Motiejunaite et al., 2020*).

From this study, it was found that there was no difference in hand grip strength between the post COVID-19 and non COVID-19 groups. This might be due to the participants' symptoms at the time of exposure to COVID-19 were mild or classified as a green patient group. This contradicted a study by *Tanrıverdi et al. (2021)* who investigated physical fitness in people who had been exposed to COVID-19 with mild and moderate symptoms,

and it was found that the group with mild symptoms had a slight decrease in muscle strength compared to the moderate group. The hand grip strength of both groups in this study was found to be moderate and there was only a 0.02 between-group difference, which was lower than the minimal clinically important difference (MCID = 0.04–0.7) (*Bohannon, 2019*).

There was a statistically significant difference in sleep quality between the post COVID-19 and non COVID-19 groups ($p < 0.05$). The non COVID-19 group had better sleep quality than the post COVID-19 group. This could be related to COVID-19 exposure, which impacted breathing difficulties and anxiety. Patients who had to be quarantined and separated from their family members could experience greater anxiety (*Souza et al., 2021*). According to a previous study, tension and depression were associated with poor sleep quality. In addition, insufficient sleep might have negative effects on the body such as fatigue, palpitations, irritability, headache (*Ingram, Maciejewski & Hand, 2020*) and stress (*Mekhael et al., 2022*).

This study found that quality of life had no significant difference between the post COVID-19 and non COVID-19 groups ($p > 0.05$). This could be because the study used a matched-pair design, where individuals with similar levels of physical activity were paired together. As a result, this pairing might lead to no noticeable differences in the quality of life between the two groups using EuroQol-5D-5L.

This study had several limitations. First, body composition can vary between genders; thus, future research should consider including only one gender to avoid this variability. Second, anxiety, depression and drug use may influence sleep quality and quality of life; therefore, further study should either collect information on or exclude cases with other factors that might affect sleep quality and quality of life. Third, this study recruited participants who had been diagnosed with COVID-19 and had recovered or completed the quarantine period within 1–3 months after being infected. Given that the post COVID-19 or long Covid condition is defined as the period extending beyond 1 month after COVID-19 infection, further study should include participants who took more than 3 months to recover or complete their quarantine period to provide a comprehensive analysis of the long-term effects of the novel coronavirus. Fourth, the non COVID-19 group was not confirmed for COVID-19 status through PCR or antibody tests in this study. There is a possibility that some asymptomatic individuals with COVID-19 were included in the control group. Future research should include verification of COVID-19 status using PCR or antibody tests to ensure the integrity of the control group. Fifth, this study did not account for other confounding factors such as chronic diseases and vaccination status when matching groups, which may impact the reliability of the comparisons made. Future studies should include these additional confounding factors when matching groups to enhance the accuracy of the results. Sixth, this study may have a small sample size. To increase the test's power, future research should be conducted with a larger sample size. Lastly, the study was a cross-sectional study that did not establish a true cause-and-effect relationship. Further studies should follow up with the participants over time to establish a causal relationship.

## CONCLUSIONS

COVID-19 impacted sleep quality and led to changes in blood oxygen saturation and breathing rate. The results of this study should be utilized for COVID-19-infected individuals' physical rehabilitation following infection. Those who have not been infected with COVID-19 should be informed of self-protection measures to avoid contracting the disease. Although a statistically significant difference was identified, it was not clinically significant for detecting exercise-induced desaturation and changes in breathing rate. Thus, these results should be used with caution.

## ACKNOWLEDGEMENTS

We would like to thank all participants for their effective cooperation in this study.

### Funding

The authors received no funding for this work.

### Competing Interests

The authors declare that they have no competing interests.

### Author Contributions

- Phatcharawadee Srirug conceived and designed the experiments, performed the experiments, analyzed the data, prepared figures and/or tables, authored or reviewed drafts of the article, and approved the final draft.
- Chatkaew Pongmala analyzed the data, prepared figures and/or tables, authored or reviewed drafts of the article, and approved the final draft.
- Balkis Mayeedeng performed the experiments, prepared figures and/or tables, authored or reviewed drafts of the article, and approved the final draft.
- Nurulfatin Yusoh performed the experiments, prepared figures and/or tables, and approved the final draft.
- Sofiya Malee performed the experiments, prepared figures and/or tables, and approved the final draft.
- Chadayu Udom conceived and designed the experiments, performed the experiments, analyzed the data, prepared figures and/or tables, authored or reviewed drafts of the article, and approved the final draft.

### Human Ethics

The following information was supplied relating to ethical approvals (*i.e.*, approving body and any reference numbers):

The Walailak University granted Ethical approval to carry out the study within its facilities (Ethical Application Ref: WUEC-22-204-01).

## Data Availability

The raw data is available in the Supplemental File.

## Supplemental Information

Supplemental information for this article can be found online at http://dx.doi.org/10.7717/peerj.18074#supplemental-information.

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
