# Peer review of "Effect of post COVID-19 on body composition, physical fitness, sleep quality and quality of life among young adults: a cross-sectional study of matched pairs"

_PeerJ, doi:10.7717/peerj.18074_

## Round 0.1 · original submission · Major Revisions

The present study is a cross-sectional study which was exploring the influence of COVID on Thai young people's physical functions and life quality. The research compared the two groups of participants with one having the history of COVID-19 while the second cohort did not have any records of the infection. The investigation was done on a number of parameters, such as body mass index, fitness, sleep pattern, and overall quality of life.

Key findings showed that while there were no meaningful differences in quality of life and body composition, but differences in oxygen saturation and respiratory rate post-exercise along with poor quality of sleep were reported.

The study design presented unambiguous interpretation particularly in the categorization of the infected persons. The uncertainty about the verification of the COVID-19 status with PCR or antibody testing, as well as the high prevalence of people that are asymptomatic, may compromise the controls, being a limitation that the authors should recognize.

An additional critique was directed at the matching algorithm that was mainly based on age, gender, and exercise levels, ignoring other confounders such as chronic disease and vaccination status. The possible bias or residual confounding that may have been introduced into the comparison may affect the reliability of the comparisons.

There are other issues regarding precision of the manuscript's language and presentation.

Overall, the study has made mention of knowledgeable outcomes in post-COVID condition among Thai university students but the reliability of the data is however somewhat constrained by the small sample size and methodological limitations. Even though, it is undisputed that it contributes meaningfully to the existing body of information, delineating on the fact that there is need to carry out further research to give comprehensive analysis of the long term effects of the novel coronavirus. The authors should turn the attention to these as well by conducting future researches in order to add more depth and value to their results.

Reviewer 1 ·

Basic reporting

This cross-sectional study aimed to assess the impact of COVID-19 on physical function and quality of life among Thai young adults. Two matched groups, one with post-COVID-19 infection and the other without, were compared in terms of body composition, physical fitness, sleep quality, and quality of life. Results indicated significant differences between the two groups in changes in oxygen saturation and respiratory rate following physical fitness tests, as well as sleep quality. However, no significant differences were observed in quality of life or body composition. The findings suggest that COVID-19 affects oxygen saturation, respiratory rate, and sleep quality in young adults. It underscores the importance of physical rehabilitation for those recovering from COVID-19 and emphasizes the need for preventive measures among those yet to contract the virus. Nonetheless, I would like to suggest some revisions to enhance the quality of your work.

Experimental design

The distinction between the post COVID-19 group and the non-COVID-19 group lacks clarity regarding how their infection statuses were confirmed (lines 99-106). Were infected individuals confirmed by PCR tests? Were non-infected individuals confirmed by serology tests? Additionally, the fact that 92 out of 128 individuals were asymptomatic infections (line 151) raises concerns that controls may have had exposure without diagnosis due to asymptomatic infection. If serology tests were not performed, it should be acknowledged as a limitation.

Further elucidation is necessary concerning the case-control matching algorithm (lines 105-106). Specifically, it is imperative to understand the rationale behind the selection of the optimal matching algorithm, including the choice of distance metric, ratio, and consideration of replacement. Such details are crucial as they guide the decision-making process regarding the trade-off between potential bias due to substantial sample loss and residual confounding resulting from poorly matched subjects.

Insufficient consideration is given to confounding factors in the case-control matching algorithm. Vital confounding factors such as comorbidities (e.g., immunodeficiency status and asthma), sleep duration within the past year, and number of COVID-19 vaccination doses should be accounted for to ensure comparability between cases and controls. However, the case-control matching were conducted solely based on age, sex, and physical activity level (lines 105-106), failing to adequately address stratified risks posed by these factors.

The method section lacks clarity and organization. Providing subtitles for lines 86-133 would enhance readability and comprehension.

Validity of the findings

The sample size calculation lacks convincing rationale. With multiple outcomes, it is unclear which effect size was utilized for the calculation. Reference to prior evidence supporting the chosen effect size is necessary. Furthermore, the study's validity is compromised by the small sample size, a limitation that should be acknowledged.

Clarification is needed regarding how visceral fat levels were determined and the unit of measurement. This information should be clearly defined in the methodology.

Definitions for "good/poor sleep quality" (Table 5) using the Pittsburgh Sleep Quality Questionnaire tool and "Not perfect/perfect health" (Table 6) using the EuroQol-5D-5L tool should be provided to ensure consistency and clarity.

The discussion concerning body composition comparison (lines 197-204) is unclear. Cross-sectional data cannot provide insight into changes in body composition in relation to infection. Similarly, conclusions regarding the deterioration of quality of life (lines 265-266) cannot be drawn from the available data. These statements should be revisited and clarified throughout the discussion.

The clinical implications of differences in respiratory rate and changes in blood oxygen saturation after the 1-minute sit-to-stand test should be discussed to strengthen the argument. While statistical significance is noted (lines 208, 223), providing the clinical significance of these differences would enhance the discussion.

For limitation 1, conducting a sensitivity analysis among females and males only is suggested. Although the sample size may be further compromised, this analysis would provide additional insights compared to mixed data from both men and women.

Reviewer 2 ·

Basic reporting

At least the title is not smart. Coronavirus 2019 appears twice in the title and needs a preposition in front of Thai young adults. Body composition and quality of life were not affected.

Overall, the application should be checked for grammatical errors.

Experimental design

The average number of days since COVID-19 infection should be noted.

SpO2 and RR should be described in Methods, including the method of calculating the difference, if the results are presented as a difference.

Are there any differences in SPO2, RR, and other pre-exercise data between the two groups?

Validity of the findings

The difference between the two groups is one time for RR and about 0.3% for SpO2. Is this difference really a meaningful difference?

Additional comments

Line52-56
Can you update it with the latest information?

Line224
RR compares the difference before and after exercise, so it cannot be said that the RR is high.

Full spelling of abbreviations is required in the footnotes of the table.

Are the blood pressure values in Table 2 correct?

Reviewer 3 ·

Basic reporting

The article meets the standard professional English.

Experimental design

The investigation in this article covers the group of post-COVID patients compared to controls for the symptoms and factors related to health and quality of life. The study is well-defined and contains no ambiguity at large.

Validity of the findings

The study adds the information of post-COVID-related symptoms in Thai adults using methodologies corroborating the findings at large.

Additional comments

A cross-sectional study of SARS-CoV-2 impact on health and other factors of Thai young adults has been carried out by Srirug et al. The study focuses on different factors being affected by COVID-19 in COVID recovered patients and has thus taken into account different methodologies.
The study has a few changes to be included; I suggest, In the introduction, changing Coronavirus 2019 (COVID-19) to “Coronavirus disease 2019”.
In lines 51-52, I suggest the authors change “has increased” to “Increased” as, only it was declared an epidemic after that.
In Line 54 remove “And” before Thailand

In line 67: “Remnant after treatment,” the authors can give exact references and explain it in more detail for clarity.

In Lines 99-106: It would be helpful for the study to mention if the groups were tested or confirmed for the presence of the SARS-CoV-2 antibodies and, thus, previous infection.

The authors can also include, if done or identified, which strain/s of SARS-CoV-2 infected the groups as it may affect the outcomes of the result.


In Lines 162-163: The information is not clearly stated in this sentence; please rephrase it.

In Table 2 change “Hospitel” to Hospital and in discussion subheadings shall be avoided

I appreciate the work of the authors and identifying the potential limitations; further follow-up studies will certainly help in bringing the broader picture of post-COVID symptoms. The study helps add information to the existing pool of post-COVID symptoms using proper methodologies.

---

## Round 0.2 · Major Revisions

Dear Authors,

Before we make a final decision on your manuscript, there are a few methodological details of your study, which need to be clarified. Particularly, we have observed that after establishing a dropout rate of 25%, you assume a total subject count per group to be 36. While the method for estimating sample size may be deemed appropriate, there are several points that need to be elucidated.

Initially, it is noted that sample size estimation has been made for COVID-19 patients and those without COVID-19. Still, it is not entirely clear which variable was used for this estimation, although the existence of an effect size is described further on. This omission necessitates clarification.

Type of statistical test has not been pointed out with which the estimation has been derived out. The identification of the test used is essential to evaluating the reliability of your sample size calculation as well as its broader applicability.

Finally, you hypothesize an effect size of 0, 51 which in your view is satisfactory enough. However, it is not clear to which of the many variables that you have identified, this effect size relates to. Does this effect size pertain to all measured variables? Could you confirm whether it was anticipated that all variables measured would exhibit an effect size of 0.51 between COVID-19 and non-COVID-19 patients?

Furthermore, in the statistical analysis section, you state, "For normal distribution, all variables were employed using the Shapiro-Wilk test." This statement is somewhat ambiguous. We can assume that you are referring to the use of the Shapiro-Wilk test to determine the normality of the variables, and based on the results, the decision to use parametric or non-parametric tests was made. However, the current wording is unclear and requires revision for better comprehension.

Once the authors clarify their method for determining the sample size, it will be necessary to identify whether the non-significant results obtained in this study may be subject to a potential Type II error.

Minor: The wording of the discussion section makes excessive reference to results already published in the study. The authors might consider rewriting part of this section to avoid redundancy. Additionally, the authors should exercise greater caution in their conclusion, given the limited sample size and the simplicity of the statistical analysis performed.

Your prompt and detailed clarification on these points will greatly assist in the evaluation of your manuscript.

Thank you for your cooperation.

Best regards,

Prof. Moran.

Reviewer 1 ·

Basic reporting

I have reviewed the revised manuscript and am pleased to see that the authors have addressed most of the comments and concerns raised in the previous review. The revisions have significantly improved the clarity and quality of the paper. I believe the current version meets the standards for publication and recommend that the manuscript be accepted. Thank you to the authors for their diligent efforts in revising their work.

Experimental design

NA

Validity of the findings

NA

Reviewer 2 ·

Basic reporting

The author responded appropriately and effectively to my comments, which I believe has improved the overall quality of the research paper.

Experimental design

None.

Validity of the findings

None.

Additional comments

None.

---

## Round 0.3 · accepted · Accept

After careful consideration of the revised manuscript and based on the positive feedback from the reviewers, I am pleased to inform you that your manuscript has been accepted for publication. The reviewers has noted that you have successfully addressed the majority of the comments and concerns raised in the previous review, significantly improving the clarity and quality of the paper. The current version of your manuscript meets the publication standards, and we appreciate your diligent efforts in revising your work.